# Risk of Neck Hematoma Following Thyroidectomy in Patients Taking Direct Oral Anticoagulants: A Propensity Score Matching Analysis from Nine High-Volume European Centers (RAGNO Study)

**DOI:** 10.3390/jcm14103435

**Published:** 2025-05-14

**Authors:** Gian Luigi Canu, Fabio Medas, Federico Cappellacci, Giulia Lanzolla, Leonardo Rossi, Francesco Pennestrì, Giacomo Di Filippo, Angeliki Chorti, Pierpaolo Gallucci, Andrea De Palma, Carlo Enrico Ambrosini, Ioannis Pliakos, Moysis Moysidis, Valentine Luzuy-Guarnero, Benoit Bédat, Giulia Salvi, Serena Elisa Tempera, Giulia Carnassale, Amelia Mattia, Giovanni Lazzari, Tommaso Guagni, Martina Izzo, Francesco Boi, Eleonora Morelli, Francesco Feroci, Emanuela Traini, Pietro Princi, Marco Stefano Demarchi, Theodosios Papavramidis, Marco Raffaelli, Gabriele Materazzi, Carmela De Crea, Pietro Giorgio Calò

**Affiliations:** 1Department of Surgical Sciences, University of Cagliari, 09042 Monserrato, Italy; fabio.medas@unica.it (F.M.); fedcapp94@gmail.com (F.C.); pgcalo@unica.it (P.G.C.); 2Endocrinology Unit, Department of Medical Sciences, University of Cagliari, 09042 Monserrato, Italy; giulia.lanzolla@unica.it (G.L.); francesco.boi@unica.it (F.B.); 3Endocrine Surgery Unit, University Hospital of Pisa, 56100 Pisa, Italy; leonardo.rossi@med.unipi.it (L.R.); depalma.and@gmail.com (A.D.P.); carloeambrosini@gmail.com (C.E.A.); gabriele.materazzi@unipi.it (G.M.); 4UOC di Chirurgia Endocrina e Metabolica, Fondazione Policlinico Universitario Agostino Gemelli IRCCS, 00168 Rome, Italy; francesco.pennestri@policlinicogemelli.it (F.P.); pierpaolo.gallucci@policlinicogemelli.it (P.G.); marco.raffaelli@unicatt.it (M.R.); carmela.decrea@unicatt.it (C.D.C.); 5Centro di Ricerca in Chirurgia delle Ghiandole Endocrine e dell’Obesità, Università Cattolica del Sacro Cuore, 00168 Rome, Italy; 6Endocrine Surgery Unit, Department of Surgery and Oncology, Verona University Hospital, 37124 Verona, Italy; giacomo.difilippo@aovr.veneto.it (G.D.F.); giovanni.lazzari@aovr.veneto.it (G.L.); eleonora.morelli@aovr.veneto.it (E.M.); 7First Propedeutic Department of Surgery, Aristotle University of Thessaloniki, AHEPA University Hospital, 546 36 Thessaloniki, Greece; chorange2404@gmail.com (A.C.); plliakos@hotmail.com (I.P.); moisisdoc@gmail.com (M.M.); papavramidis@hotmail.com (T.P.); 8Unit of Minimally Invasive Surgery, Euromedica Kyanous Stavros, 546 36 Thessaloniki, Greece; 9Department of Thoracic and Endocrine Surgery and Faculty of Medicine, University Hospitals of Geneva, 1205 Geneva, Switzerland; valentine.luzuy-guarnero@hug.ch (V.L.-G.); benoit.bedat@hug.ch (B.B.); marcostefano.demarchi@hug.ch (M.S.D.); 10Multifunctional Center of Endocrine Surgery, Cristo Re Hospital, 00167 Rome, Italy; giulia.salvi2991@gmail.com (G.S.); serena.tempera@gmail.com (S.E.T.); pietroprinci@yahoo.it (P.P.); 11Endocrine Surgery Unity, San Carlo di Nancy Hospital—GVM Care and Research, 00165 Rome, Italy; gcarnassale@gvmnet.it (G.C.); amattia@gvmnet.it (A.M.); etraini@gvmnet.it (E.T.); 12Department of General Surgery, SS Cosma e Damiano Hospital, 51017 Pescia, Italy; tommaso.guagni@uslcentro.toscana.it (T.G.); martina.izzo@uslcentro.toscana.it (M.I.); francesco.feroci@uslcentro.toscana.it (F.F.); 13Department of General and Oncologic Surgery, Santo Stefano Hospital, 59100 Prato, Italy; 14Unicamillus International Medical University, 00131 Rome, Italy

**Keywords:** thyroid surgery, thyroidectomy, neck hematoma, bleeding, hemorrhage, complications, anticoagulants, direct oral anticoagulants, DOACs

## Abstract

**Background:** Postoperative neck hematoma is an infrequent but potentially fatal complication following thyroidectomy. The main aim of this study was to evaluate the impact of direct oral anticoagulants (DOACs) on the occurrence of this complication. **Methods:** Patients who underwent thyroidectomy between January 2020 and December 2022 in nine high-volume thyroid surgery centers in Europe were retrospectively evaluated. Based on taking direct oral anticoagulants, patients were divided into two groups: the DOAC Group and the Control Group. Propensity score matching 1:1 was performed between the two groups, which were then compared through a univariate analysis. **Results:** The total number of patients enrolled based on the inclusion/exclusion criteria was 8985. Following propensity score matching, the study population consisted of 316 patients: 158 in the DOAC Group and 158 in the Control Group. In the DOAC Group, the overall incidence of neck hematoma was 5.70% (4.43% for neck hematomas managed conservatively, and 1.27% for those that required surgical revision of hemostasis). No statistically significant difference was found between the two groups in terms of the incidence of this complication. Hospital readmission due to neck hematoma was not observed in any patient. No statistically significant difference was found between the two groups in terms of the timing of the onset of neck hematomas that required surgical revision of hemostasis. **Conclusions:** This study showed that direct oral anticoagulants, in the field of thyroid surgery, have no impact on the occurrence of postoperative neck hematoma. Therefore, based on our findings, it can be concluded that thyroidectomy can be safely performed in patients taking this class of drugs.

## 1. Introduction

Postoperative neck hematoma is an infrequent but potentially fatal complication following thyroidectomy. Its occurrence rate, according to what is reported in the literature, reaches up to 6.5% [1,2,3,4,5,6,7,8,9,10,11,12,13,14,15,16,17,18,19].

This complication can lead to acute airway obstruction; therefore, if its onset is not managed promptly and adequately, serious neurological complications or even death of the patient may occur. Neck hematoma with airway compromise requires immediate surgical revision of hemostasis. However, it is important to emphasize that this complication, in most cases, can be managed conservatively [1,2,3,4,5,6,7,8,9,10,11,12,13,14,15,16,17,18,19].

In terms of timing of onset, neck hematoma generally occurs within 24 h after surgery, particularly in the first 6 h, and is quite rare after 24 h [1,6,7,8,12,13,15,17,18,19].

Since the occurrence of this complication is not common, accurate statistical evaluation of the risk factors is challenging. So far, various risk factors for the onset of neck hematoma have been described in the literature: male gender, older age, high blood pressure, higher body mass index, active smoking, postoperative vomiting and coughing, extent of the operation (bilateral thyroid surgery, neck dissection), previous thyroidectomy, low-volume hospitals, low surgeon experience, retrosternal goiter, large thyroid gland, and Graves’ disease [1,2,3,4,5,6,7,8,9,10,11,12,13,14,15,16,17,18,19].

As for the impact of antithrombotic drugs, namely antiplatelet agents and anticoagulants, on the occurrence of this complication, studies in the literature report discordant results [6,7,8,9,10,11,12,13,14,15,16,17,18,19]. Moreover, in various analyses, antiplatelet agents and anticoagulants are grouped together and considered as a single variable [6,8,9,10]. Furthermore, with regard to anticoagulants, and specifically direct oral anticoagulants (DOACs), to our knowledge, there are only two studies in the literature that take them into consideration [7,19]. However, in the study by Oltmann et al. [19] DOACs were grouped together with traditional oral anticoagulants (vitamin K antagonists), while in the other study, which is our previous multicenter analysis on risk factors for neck hematoma after thyroid surgery, DOACs were considered but did not represent the main subject of the investigation [7].

Nowadays, direct oral anticoagulants, namely apixaban, rivaroxaban, edoxaban, and dabigatran, are widely utilized drugs. They exert their effect by means of the inhibition of factor Xa (apixaban, rivaroxaban, and edoxaban) or factor IIa (dabigatran) [20,21,22].

DOACs are currently the drugs of choice for the prevention of stroke in patients with atrial fibrillation and for the prevention as well as treatment of venous thromboembolism [20,21,22].

Unlike traditional oral anticoagulants, such as warfarin and acenocoumarol, DOACs have a lower risk of bleeding, a stable pharmacokinetic profile that allows for standardized dosing with no need for routine monitoring, and fewer potential interactions with other drugs [20,21,22].

In light of these advantages, since their introduction to the market between 2008 and 2015, the prescription of these drugs has progressively increased over the years. Nowadays, in many countries, DOACs have surpassed vitamin K antagonists, becoming the most commonly utilized method of oral anticoagulation. In this regard, it is important to emphasize that about 20% of patients treated with DOACs undergo an elective or urgent procedure every year [20,21,22].

The main aim of this study was to assess the impact of direct oral anticoagulants on the occurrence of postoperative neck hematoma following thyroidectomy.

## 2. Materials and Methods

### 2.1. Study Design

This is a retrospective, multicenter, international study on patients submitted to thyroid surgery between January 2020 and December 2022.

Data analyzed come from nine high-volume thyroid surgery centers (seven in Italy, one in Switzerland, and one in Greece):-Unit of General Surgery, University Hospital of Cagliari, Monserrato (CA), Italy;-Unit of Endocrine Surgery, University Hospital of Pisa, Pisa, Italy;-Unit of Endocrine and Metabolic Surgery, “Fondazione Policlinico Universitario A. Gemelli IRCCS”, Rome, Italy;-Division of Thoracic and Endocrine Surgery, Geneva University Hospitals, Geneva, Switzerland;-Unit of Minimally Invasive Surgery, Euromedica Kyanous Stavros, Thessaloniki, Greece;-Unit of General Surgery, “Santo Stefano” Hospital, Prato, Italy;-Unit of Endocrine Surgery, University Hospital of Verona, Verona, Italy;-Multifunctional Center of Endocrine Surgery, “Cristo Re” Hospital, Rome, Italy;-Unit of Endocrine Surgery, “San Carlo di Nancy” Hospital—GVM Care and Research, Rome, Italy.

Patients who underwent conventional open thyroidectomy, including those simultaneously submitted to neck dissection or parathyroidectomy, were enrolled in this investigation.

As exclusion criteria, the following were considered: age < 18 years, patients with coagulation disorders (involving platelets and/or coagulation factors), patients taking antiplatelet agents or anticoagulants different from DOACs, and incomplete data.

Based on whether they took direct oral anticoagulants, patients were divided into two groups: the DOAC Group and the Control Group.

Demographic and preoperative features, details of the surgical procedure, postoperative hospital stay, histopathologic findings, and complications were evaluated.

### 2.2. Endpoints

The primary endpoint was to evaluate the impact of direct oral anticoagulants on the occurrence of postoperative neck hematoma. For this purpose, the incidence of this complication (overall, for neck hematomas managed conservatively, and for those that required surgical revision of hemostasis), the rate of hospital readmission due to its occurrence, and the timing of the onset of hematomas that required surgical revision of hemostasis were assessed.

As secondary endpoints, the use of drains, operative time, postoperative hospital stay, and the other early complications of thyroidectomy (hypoparathyroidism, recurrent laryngeal nerve lesion, and wound infection) were investigated.

### 2.3. Perioperative Management of Direct Oral Anticoagulants

Direct oral anticoagulants were discontinued 48–72 h before surgery (in consideration of renal function) and reintroduced 24–48 h after the operation (based on the center’s practices), without using bridging anticoagulation.

### 2.4. Surgical Procedure

All thyroidectomies (which included total thyroidectomies, hemithyroidectomies, and completion thyroidectomies) were performed via a conventional open approach.

Intraoperative nerve monitoring (IONM), energy-based devices, and topical hemostatic agents were utilized on the basis of the decision of the operating surgeon or depending on their availability.

Drains were used at the discretion of the surgeons.

The operative time (from skin incision to skin closure) was reported in minutes.

### 2.5. Evaluation of Complications

Postoperative neck hematomas were categorized based on the need for the surgical revision of hemostasis. Regarding the timing of the onset of neck hematomas that required surgical revision of hemostasis (evaluated from the end of the operation to the diagnosis of the complication), three time intervals were distinguished: within 6 h after the end of the surgical procedure, between 7 and 24 h, and after 24 h.

Postoperative hypoparathyroidism was defined based on iPTH values after the surgical procedure (normal range: 10–65 pg/mL).

A recurrent laryngeal nerve lesion was verified by means of postoperative fibrolaryngoscopy. Following surgery, this examination was performed in the event of a loss of signal at intraoperative nerve monitoring or postoperative hoarseness.

### 2.6. Statistical Analysis

Statistical analysis was conducted using IBM SPSS Statistic^®®^ (version 30.0.0.0).

Propensity score matching 1:1 was performed between the two groups. The match tolerance was set at 0.050. “Without replacement” and “Randomize case order when drawing matches” functions were applied. The variables included in this statistical technique were the following: gender, high blood pressure, hyperthyroidism, retrosternal goiter, surgical procedure, volume of surgeons, hemostasis technique, IONM, topical hemostatic agents, and Graves’ disease.

The two groups were compared through univariate analysis. The chi-squared test or Fisher’s exact test were used for categorical variables. The presence of a normal distribution of continuous variables was evaluated by means of the Shapiro–Wilks test. On the basis of the results obtained through the latter test, continuous variables, reported as the median and interquartile range (IQR), were analyzed using the Mann–Whitney U test.

The threshold for statistical significance was set at *p* < 0.05.

## 3. Results

There were a total of 8985 patients enrolled based on the inclusion/exclusion criteria: 158 in the DOAC Group and 8827 in the Control Group. Following propensity score matching, the study population consisted of 316 patients: 158 in the DOAC Group and 158 in the Control Group.

The flowchart regarding the study population is shown in Figure 1.

### 3.1. Baseline Features of the Study Population

A statistically significant difference was found between the two groups in terms of age. The median age in the DOAC Group (70, IQR: 64–76, years) was significantly higher than in the Control Group (58, IQR: 49–65, years) (*p* < 0.001).

The other variables were comparable.

Complete results are reported in Table 1.

### 3.2. Occurrence of Postoperative Neck Hematoma

The overall incidence of neck hematoma in the study population was 5.06% (3.48% for neck hematomas managed conservatively, and 1.58% for those that required surgical revision of hemostasis).

In the DOAC Group, the overall incidence of neck hematoma was 5.70% (4.43% for neck hematomas managed conservatively, and 1.27% for those that required surgical revision of hemostasis).

In the Control Group, the overall incidence of neck hematoma was 4.43% (2.53% for neck hematomas managed conservatively, and 1.90% for those that required surgical revision of hemostasis).

No statistically significant difference was found between the two groups in terms of the incidence of this complication (Table 2).

Hospital readmission due to this complication was not observed in any patient.

The timing of the onset of neck hematomas that required surgical revision of hemostasis in the study population was within 6 h after the end of the surgical procedure in two (40%) patients, and between 7 and 24 h in three (60%). No patient developed this complication after 24 h from the end of surgery.

In the DOAC Group, the timing of the onset of neck hematomas that required surgical revision of hemostasis was within 6 h after the end of the surgical procedure in one (50%) patient, and between 7 and 24 h in one (50%).

In the Control Group, the timing of the onset of neck hematomas that required surgical revision of hemostasis was within 6 h after the end of the surgical procedure in one (33.33%) patient, and between 7 and 24 h in two (66.67%).

No statistically significant difference was found between the two groups in terms of the timing of the onset of neck hematomas that required surgical revision of hemostasis (*p* = 1.000).

### 3.3. Secondary Endpoints

No statistically significant difference was found between the two groups in terms of the use of drains, operative time, postoperative hospital stay, postoperative hypoparathyroidism, recurrent laryngeal nerve injury, and wound infection.

The complete results are reported in Table 3.

## 4. Discussion

To the best of our knowledge, this study is the first with the specific aim of evaluating the impact of direct oral anticoagulants, considered singularly, on the occurrence of neck hematoma following thyroid surgery [1,2,3,4,5,6,7,8,9,10,11,12,13,14,15,16,17,18,19].

In the case of elective surgery, it is necessary to discontinue these drugs in the perioperative period, without, however, the need for bridging anticoagulation. The timing of the interruption and reintroduction is based mainly on the bleeding risk associated with the operation, but also on the type of anatomical area involved in the surgical procedure. In fact, regarding this last point, it is important to emphasize that bleeding after operations involving closed anatomical spaces, even if the risk of developing this complication is not considered high, as in the case of thyroidectomy, can lead to very serious consequences. Regarding the interruption, it is also necessary to take into account the patient’s renal function [20].

In patients of our study, DOACs were discontinued 48–72 h before surgery (in consideration of renal function) and reintroduced 24–48 h after the operation (based on the center’s practices).

In order to evaluate the impact of direct oral anticoagulants on the occurrence of neck hematoma following thyroidectomy, which represents the primary endpoint of this study, an analysis was conducted on the incidence of this complication (overall, for neck hematomas managed conservatively, and for those that required surgical revision of hemostasis), the rate of hospital readmission due to its occurrence, and the timing of the onset of hematomas that required surgical revision of hemostasis. Furthermore, as secondary endpoints, the use of drains, operative time, postoperative hospital stay, and other early complications of thyroid surgery were investigated.

In order to balance the two groups analyzed, both from a numerical point of view and with regard to the baseline features, propensity score matching (1:1) was performed. The variables included in this statistical technique were the following: gender, high blood pressure, hyperthyroidism, retrosternal goiter, surgical procedure, volume of activity of the surgeons, hemostasis technique, intraoperative neuromonitoring, topical hemostatic agents, and Graves’ disease. These variables were included in consideration of their possible impact on the endpoints of the study [1,2,3,4,5,6,7,8,9,10,11,12,13,14,15,16,17,18,19]. Differently, age was not included despite having been reported by some authors as a risk factor for postoperative neck hematoma [7,8,9,10,14]. This variable, in the comparison between the two groups, was significantly higher in patients treated with DOACs. This result can be attributed to the fact that therapy with direct oral anticoagulants is more commonly taken by older patients. This variable was not included in the propensity score matching in order not to exclude from the study population, and in particular from the Control Group, a very relevant category of patients who undergo thyroid surgery, namely that consisting of younger individuals. This methodological choice, in our opinion, makes the obtained results more generalizable.

In our analysis, in the study population, the overall incidence of neck hematoma was 5.06% (3.48% for neck hematomas managed conservatively, and 1.58% for those that required surgical revision of hemostasis). In patients treated with DOACs, the overall incidence of neck hematoma was 5.70% (4.43% for neck hematomas managed conservatively, and 1.27% for those that required surgical revision of hemostasis). Regarding incidence rates, no statistically significant difference was found between the two groups. It is important to emphasize that this result was obtained even though patients in therapy with DOACs were significantly older, and therefore, according to what has been described by some authors, at greater risk for developing postoperative neck hematoma.

With regard to hospital readmission due to this complication, it was not observed in any patient in the study population.

The timing of the onset of neck hematomas that required surgical revision of hemostasis in the study population was within 6 h of the end of surgery in two (40%) patients, and between 7 and 24 h in three (60%). No patient developed this complication 24 h after the end of the surgical procedure. In patients treated with DOACs, the timing of the onset of neck hematomas that required surgical revision of hemostasis was within 6 h after the end of the surgical procedure in one (50%) patient, and between 7 and 24 h in one (50%). Also regarding this result, no statistically significant difference was found between the two groups.

Our findings about the incidence rates of this complication are comparable to those reported by other authors [1,2,3,4,5,6,7,8,9,10,11,12,13,14,15,16,17,18,19]. With regard to the timing of the onset of neck hematomas that required the surgical revision of hemostasis, there was no prevalence of occurrence in the first six hours, as described in other studies. This result was very probably influenced by the small number of cases examined. However, it is important to emphasize that in all patients this complication developed within 24 h [1,6,7,8,12,13,15,17,18,19].

Concerning the use of drains, operative time, postoperative hospital stay, and the other early complications of thyroidectomy, namely the secondary endpoints, the comparison between the two groups, also in this case, showed no statistically significant differences. The results regarding operative times, but also complications, reveal that there were no particular additional intraoperative difficulties in patients treated with DOACs.

The main limitation of this study is the retrospective nature of our analysis, therefore putting it at risk of bias. However, in this regard, it is important to emphasize that the application of propensity score matching, through which the two groups were balanced, has certainly limited bias, giving greater value to the results obtained. There are also two other limitations, again related to the retrospective nature of the study. The first is that, for neck hematomas managed conservatively, it was not possible to assess the timing of the onset, as this information was not available in most cases. The other limitation is represented by the fact that it was not possible to retrieve which drug belonging to the class of DOACs was taken by the patients included in our study. For this reason, it was not possible to assess the specific impact of each of the different DOACs on the occurrence of neck hematoma.

## 5. Conclusions

This study showed that direct oral anticoagulants, in the field of thyroid surgery, have no impact on the occurrence of postoperative neck hematoma, nor on other surgical outcomes. Therefore, based on our findings, it can be concluded that thyroidectomy, in patients taking this class of drugs, can be safely performed and without any particular additional technical difficulties.

However, considering the limitations of our investigation, further studies, preferably prospective and with larger populations, are needed to better examine this subject.

## Figures and Tables

**Figure 1 jcm-14-03435-f001:**
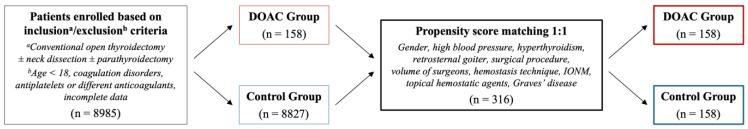
Flowchart regarding the study population.

**Table 1 jcm-14-03435-t001:** Baseline features of the study population.

	Total	DOAC Group	Control Group	*p*-Value
	(n = 316)	(n = 158)	(n = 158)	
Gender				
Male	161(50.95%)	81 (51.27%)	80 (50.63%)	0.91
Female	155 (49.05%)	77 (48.73%)	78 (49.37%)	
Age	64.50 (55–72)	70 (64–76)	58 (49–65)	< 0.001
(years, median and IQR)				
High blood pressure	209 (66.14%)	108 (68.35%)	101 (63.92%)	0.405
Hyperthyroidism	124 (39.24%)	64 (40.51%)	60 (37.97%)	0.645
Retrosternal goiter	43 (13.61%)	21 (13.29%)	22 (13.92%)	0.87
Type of Thyroidectomy				
TT	254 (80.38%)	126 (79.75%)	128 (81.01%)	0.596
HT	53 (16.77%)	26 (16.45%)	27 (17.09%)	
CT	9 (2.85%)	6 (3.80%)	3 (1.90%)	
Central Neck Dissection				
No	301 (95.25%)	149 (94.31%)	152 (96.20%)	0.518
Unilateral	1 (0.32%)	1 (0.63%)	0	
Bilateral	14 (4.43%)	8 (5.06%)	6 (3.80%)	
Lateral Neck Dissection				
No	309 (97.78%)	153 (96.84%)	156 (98.73%)	0.448
Unilateral	7 (2.22%)	5 (3.16%)	2 (1.27%)	
Parathyroidectomy	5 (1.58%)	3 (1.90%)	2 (1.27%)	1
Volume of Surgeons *				
<25	30 (9.49%)	13 (8.23%)	17 (10.76%)	0.623
25–50	10 (3.16%)	6 (3.80%)	4 (2.53%)	
>50	276 (87.35%)	139 (87.97%)	137 (86.71%)	
Hemostasis Technique				
Conventional	51 (16.14%)	29 (18.35%)	22 (13.92%)	0.214
Advanced bipolar EBD	149 (47.15%)	79 (50.00%)	70 (44.31%)	
Ultrasonic EBD	99 (31.33%)	41 (25.95%)	58 (36.71%)	
Hybrid EBD	17 (5.38%)	9 (5.70%)	8 (5.06%)	
Use of IONM				
No	152 (48.10%)	81 (51.27%)	71 (44.94%)	0.395
Intermittent IONM	138 (43.67%)	63 (39.87%)	75 (47.47%)	
Continuous IONM	26 (8.23%)	14 (8.86%)	12 (7.59%)	
Use of hemostatic agents	96 (30.38%)	51 (32.28%)	45 (28.48%)	0.463
Histological Diagnosis				
Graves’ disease	41 (12.97%)	19 (12.03%)	22 (13.92%)	0.616
Hashimoto’s thyroiditis	63 (19.94%)	30 (18.99%)	33 (20.89%)	0.673
Malignancy	114 (36.08%)	57 (36.08%)	57 (36.08%)	1

IQR, interquartile range; TT, total thyroidectomy; HT, hemithyroidectomy; CT, completion thyroidectomy; EBD, energy-based device; and IONM, intraoperative nerve monitoring. * Thyroidectomies per year.

**Table 2 jcm-14-03435-t002:** Occurrence of postoperative neck hematoma.

	Total	DOAC Group	Control Group	*p*-Value
	(n = 316)	(n = 158)	(n = 158)	
Neck hematoma				
Overall	16 (5.06%)	9 (5.70%)	7 (4.43%)	0.608
No surgical revision	11 (3.48%)	7 (4.43%)	4 (2.53%)	0.357
Surgical revision	5 (1.58%)	2 (1.27%)	3 (1.90%)	1
Readmission for hematoma	0	0	0	-

**Table 3 jcm-14-03435-t003:** Secondary endpoints.

	Total (n = 316)	DOAC Group (n = 158)	Control Group (n = 158)	*p*-Value
Use of drains	247 (78.16%)	127 (80.38%)	120 (75.95%)	0.341
Operative time (minutes, median and IQR)	70 (50–110)	70 (50–114.25)	70 (50–110)	0.841
Postoperative hospital stay (days, median and IQR)	2 (1–3)	2 (1–3)	2 (1–3)	0.076
Postoperative hypoparathyroidism	42 (13.29%)	21 (13.29%)	21 (13.29%)	1.000
RLN Injury Unilateral Bilateral	10 (3.16%) 1 (0.32%)	4 (2.53%) 1 (0.63%)	6 (3.80%) 0	0.520 1.000
Wound infection	3 (0.95%)	0	3 (1.90%)	0.248

IQR, interquartile range; RLN, recurrent laryngeal nerve.

## Data Availability

The data that support the findings of this study are available from the corresponding author upon reasonable request.

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
