# Peer review of "Risk of Neck Hematoma Following Thyroidectomy in Patients Taking Direct Oral Anticoagulants: A Propensity Score Matching Analysis from Nine High-Volume European Centers (RAGNO Study)"

_jcm, 2025, doi:10.3390/jcm14103435_

Round 1
Reviewer 1 Report
Comments and Suggestions for Authors
Review Report for Manuscript
"Risk of neck hematoma following thyroidectomy in patients taking direct oral anticoagulants: a propensity score matching analysis from nine high-volume European centers (RAGNO study)"
This study evaluates a large cohort of patients to assess the role of direct oral anticoagulants (DOACs) in the occurrence of neck hematoma following thyroidectomy. The introduction, methodology, results, and discussion sections are well structured.
Minor comments:
- Considering that the propensity score matching was based on variables such as gender, high blood pressure, hyperthyroidism, retrosternal goiter, surgical procedure, surgeon volume, hemostasis technique, IONM, topical hemostatic agents, and Graves’ disease, it is inappropriate to perform statistical comparisons for these parameters between the two groups in Table 1. These variables were already balanced through matching, and presenting p-values for them may be misleading.
- Lines 242–258 in the discussion section read more like introductory background rather than a discussion of the study’s findings. This part should be shortened and, where appropriate, moved to the introduction. The discussion should instead focus more on interpreting the results in light of the current literature.
Author Response
Dear Reviewer 1,
thank you for your analysis of our manuscript and for your constructive and precious suggestions.
Below are the answers, point by point, to your comments.
1. Thanks for your observation. However, in our opinion, comparing the two groups after propensity score matching is useful to confirm that the baseline features have been successfully balanced. In addition, Table 1 also includes other baseline features (such as age) that were not included in propensity score matching. We consulted several other articles and in many of them, after propensity score matching, a comparison (shown in a table) was performed.
2. I fully agree with your remark. The section in question was slightly shortened and moved to the “Introduction” section.
Best regards.
Reviewer 2 Report
Comments and Suggestions for Authors
Please add the abbreviation' DOACs' after 'direct oral anticoagulants' on line 53 (please see the uploaded file). This abbreviation appears on line 96.
In the discussion chapter, it might be helpful to make some remarks about the drugs that belong to the DOACs group, even if this medication is generally known for its risks and benefits.

Author Response
Dear Reviewer 2,
thank you for your analysis of our manuscript and for your constructive and precious suggestions.
Below are the answers, point by point, to your comments.
- Thank you for your correct observation. We have added the abbreviation (DOACs) where you suggested.
- Thank you for your valuable suggestion. In the new version of the manuscript, some details regarding DOACs (such as the different drugs belonging to this class, indications for use, advantages over other anticoagulants, market penetration, and management in case of surgery) are described at the end of the “Introduction” section and at the beginning of the “Discussion” section.
Best regards.
Reviewer 3 Report
Comments and Suggestions for Authors
The paper “Risk of neck hematoma following thyroidectomy in patients taking direct oral anticoagulants: a propensity score matching analysis from nine high-volume European centers (RAGNO study)” focuses on the relationship between direct oral anticoagulants (DOACs) taken by patients undergoing thyroid surgery and postoperative cervical hematoma. The topic selection is of clinical significance and provides a reference for clinical decision-making. The research design was reasonable. A multicenter, retrospective study and propensity score matching analysis were adopted. The statistical methods were appropriately used, and confounding factors were controlled to a certain extent. However, there are still some areas for improvement in the research to further enhance the quality of the study and the reliability of the conclusions. Here are some specific comments.
Comments:
Q1. The article mentioned that patients with incomplete data were excluded, but did not specify the exact proportion of incomplete data or the types of missing data. If the missing data has systematic bias, it may affect the accuracy of the research results.
Q2. Although some preoperative factors (such as hypertension, hyperthyroidism, etc.) were considered in the study, some comorbidities that might affect the risk of postoperative bleeding (such as diabetes, liver and kidney dysfunction, etc.) were not evaluated and analyzed in detail.
Q3. The text only mentions the time range for discontinuation and re-initiation of DOACs, but the pharmacokinetics and pharmacodynamics of different types of DOACs vary, and their impacts on the risk of surgical bleeding may differ, which was suggested to be discussed.
Q4. The control group consisted of patients who did not take DOACs, but the anticoagulation status of the patients in the control group was not detailed. If there were some patients with potential anticoagulation risks (such as taking other antiplatelet drugs, etc.) in the control group, it might interfere with the accuracy of the research results.
Q5. The research conclusion indicates that DOACs has no effect on cervical hematoma after thyroid surgery. However, in clinical practice, doctors still need to consider other factors (such as the overall bleeding risk of the patient, the risk of thrombosis, etc.) to decide whether to discontinue DOACs.
Q6. The significance of this study should be further strengthened. Why should we take anticoagulants after surgery?
Author Response
Dear Reviewer 3,
thank you for your analysis of our manuscript and for your constructive and precious suggestions.
Below are the answers, point by point, to your comments.
Q1. Thank you for your valuable observation. Incomplete data refers to missing information relating to baseline features and endpoints. In the case of missing data, even for a single variable, the patient was excluded. In this regard, we would like to emphasize that a negligible percentage of patients were excluded from the study due to incomplete data. This is due to the fact that the variables considered and analyzed were established by the participating centers based on the possibility of retrieving them (from existing internal databases or medical records).
Q2. I fully agree with your remark. However, due to the retrospective nature of our analysis, it was not possible to retrieve information on other comorbidities (such as diabetes, liver and kidney dysfunction). In this regard, we would like to emphasize that several risk factors for the occurrence of neck hematoma have been considered and balanced (through propensity score matching).
Q3. Thank you for your valuable observation. Unfortunately, due to the retrospective nature of our analysis, it was not possible to retrieve which drug belonging to the class of DOACs was taken by the patients included in our study. For this reason, it was not possible to assess the specific impact of each of the different DOACs on the occurrence of neck hematoma. This fact was included among the limitations of the study. Regarding the time range for discontinuation and re-initiation of the different DOACs, it is important to emphasize that, in clinical practice, despite some differences in pharmacokinetics and pharmacodynamics, it is quite comparable.
Q4. I fully agree with your remark. In this regard, we would like to point out that in the “Materials and Methods” section and in Figure 1, it is clearly stated that patients with coagulation disorders (involving platelets and/or coagulation factors) and patients taking antiplatelet agents or anticoagulants different from DOACs were excluded from the study.
Q5. Thank you for your observation. However, discontinuation of DOACs in the perioperative period is clearly recommended by current guidelines. As specified in the “Discussion” section, the timing of the interruption and reintroduction is based mainly on the bleeding risk associated with the operation, but also on the type of anatomical area involved in the surgical procedure. In fact, regarding this last point, it is important to emphasize that bleeding after operations involving closed anatomical spaces, even if the risk of developing this complication is not considered high, as in the case of thyroidectomy, can lead to very serious consequences. Ultimately, not discontinuing this class of drugs during the perioperative period of thyroidectomy would be too dangerous in relation to the risk of bleeding and its possible consequences.
Q6. Thank you for your observation. However, in this regard, we would like to emphasize that surgery (let alone thyroidectomy) is not an indication for the use of DOACs. As specified in the “Introduction” section, DOACs are currently the drugs of choice for the prevention of stroke in patients with atrial fibrillation and for the prevention and treatment of venous thromboembolism. Ultimately, patients take this class of drugs for reasons unrelated to surgery. Our study showed that, in the field of thyroid surgery, direct oral anticoagulants (taken by patients for other reasons) have no impact on the occurrence of postoperative neck hematoma, nor on other surgical outcomes.
Best regards.
Round 2
Reviewer 3 Report
Comments and Suggestions for Authors
I have no additional comments.
Author Response
Thank you for your evaluation.
Best regards.